# Catalytic Properties of Double Substituted Lanthanum Cobaltite Nanostructured Coatings Prepared by Reactive Magnetron Sputtering

**Mohammad Arab Pour Yazdi [1], Leonardo Lizarraga [2,3], Philippe Vernoux [2], Alain Billard [1] and Pascal Briois [1,***

[1] FEMTO-ST Institute (UMR CNRS 6174), Université Bourgogne Franche-Comté, UTBM, 2 Place Lucien Tharradin, F-25200 Montbéliard CEDEX, France; mohammad.arab-pour-yazdi@utbm.fr (M.A.P.Y.); alain.billard@utbm.fr (A.B.)

[2] Université Lyon, Université Claude Bernard Lyon 1, CNRS—IRCELYON—UMR 5256, 2 avenue A. Einstein, 69626 Villeurbanne, France; leonardo.lizarraga@cibion.conicet.gov.ar (L.L.); philippe.vernoux@ircelyon.univ-lyon1.fr (P.V.)

[3] CIBION, Centro de Investigaciones en Bionanociencias, CONICET, Consejo Nacional de Investigaciones Científicas y Técnicas CONICET, Godoy Cruz 2390, Buenos Aires C1425FQD, Argentina

* Correspondence: pascal.briois@utbm.fr; Tel.: +33-384-583-701

**Abstract:** Lanthanum perovskites are promising candidates to replace platinum group metal (PGM), especially regarding catalytic oxidation reactions. We have prepared thin catalytic coatings of Sr and Ag doped lanthanum perovskite by using the cathodic co-sputtering magnetron method in reactive condition. Such development of catalytic films may optimize the surface/bulk ratio to save raw materials, since a porous coating can combine a large exchange surface with the gas phase with an extremely low loading. The sputtering deposition process was optimized to generate crystallized and thin perovskites films on alumina substrates. We found that high Ag contents has a strong impact on the morphology of the coatings. High Ag loadings favor the growth of covering films with a porous wire-like morphology showing a good catalytic activity for CO oxidation. The most active composition displays similar catalytic performances than those of a Pt film. In addition, this porous coating is also efficient for CO and NO oxidation in a simulated Diesel exhaust gas mixture, demonstrating the promising catalytic properties of such nanostructured thin sputtered perovskite films.

**Keywords:** perovskite; catalytic coating; CO oxidation; cathodic sputtering method

## 1. Introduction

The family of perovskite oxides is known for its catalytic properties, hydrothermal stability, high recyclability and low cost compared with Platinum Group Metal (PGM) [1–3]. The general formula of perovskites is $ABO_3$ where the larger size A-cation presents a 12-coordination number and the B-cation coordinates with 6 neighboring atoms [4]. Partial substitution of A and/or B atoms with other elements showing redox properties may enhance the catalytic activity due to the generation of structural defects such as anionic or cationic vacancies and/or modification of the oxidation state of B cations to maintain the electro-neutrality [5]. Lanthanum perovskites are promising candidates to replace noble metals (Pt, Pd, etc.) [6,7], especially regarding catalytic oxidation reactions. Lanthanum cobaltite ($LaCoO_3$) is one of the most promising catalysts for the oxidation of gaseous pollutants such as carbon monoxide, unburnt hydrocarbons and nitrogen oxide [7,8]. Lanthanum cobaltites are used in many others fields due to their magnetic properties as well as their mixed ionic and electronic

conductivity [9–12]. Most of these applications require the implementation of thin films that act as electrodes for fuel cells [13], thermoelectric processes [14], sensors [15], and magnetoresistance devices [16]. These electric, magnetic and electrocatalytic properties depend on the composition but also on the microstructure and morphology of coatings, which strongly depend on the preparation method [17,18]. Several techniques are reported for the synthesis of perovskite thin films such as the ground-frost [7], casting in band [18], chemical vapor deposition (CVD) [17], spray painting [19], sol-gel [7], atomic layer deposition (ALD) [20], laser pulsed deposition (PLD) [21], and physical vapor deposition (PVD) [22,23]. Regarding catalytic applications, the development of submicrometric catalytic films may optimize the surface/bulk ratio to save raw materials. Indeed, a thin catalytic coating can combine a large exchange surface with the gas phase with an extremely low loading. This is particularly suitable for air cleaning since gaseous pollutants only lick the surface of the catalyst. In addition, thin catalytic coatings could be deposited on hot substrates such as collector walls in thermal engine exhausts for removing pollutants or of radiators for improving indoor air quality. Few studies on the development of thin perovskite coatings for catalytic applications are reported in the literature [24–26]. The challenge lies in preparing adherent, thermally stable and pure crystallized perovskite films without any secondary parasite phase [18], showing appropriate compositions and nanostructures for catalysis [7,8,16,18]. The porosity and the specific surface areas have to be optimized to counter-balance the low quantity of materials involved in submicrometric catalytic coatings. In the present study, we used the cathodic co-sputtering magnetron method in reactive condition to prepare submicrometric nanostructured coatings of perovskites. This technique allows the deposition of pure and adherent coatings of complex oxides with a controlled and reproducible manner, while respecting the environment [27,28]. We have chosen to prepare Sr and Ag-doped lanthanum cobaltite as this perovskite composition is one of the most active for CO oxidation [8]. The partial substitution of $La^{3+}$ cations by $Sr^{2+}$ ones improves the thermal stability and then the specific surface area of pure $LaCoO_3$ [29] and also enhances the surface concentration of oxygen vacancies involved in the oxidation catalytic mechanism [30]. The partial substitution of $La^{3+}$ by $Ag^+$ can also increase the catalytic properties of lanthanum cobaltites for oxidation reactions due to the formation of oxygen vacancies [31] and the stabilisation of Ag nanoparticles on the oxide surface [6,32]. The different parameters (intensities of current applied on the metallic targets, total pressure in the chamber, oxygen partial pressure, etc.) of the reactive magnetron sputtering preparation method were tuned to achieve pure and adherent coatings of $LaCoO_3$ on alumina dense membranes. After a calcination at 500 °C, crystallized and dense cubic perovskite films of around 1.5 μm thick were achieved with the targeted La/Co stoichiometry. Different compositions of $La_{1-x-y}Sr_xAg_yCoO_{3-\alpha}$ ($x = 0.13–0.28$, $y = 0.14–0.48$) doped perovskites were sputtered on alumina disks. We found that the incorporation of high contents of Ag can strongly modify the morphology of the coatings, increasing their porosity. The catalytic performance of the perovskite catalytic coatings for CO oxidation was found to be improved by the double substitution of Sr and Ag. The most active composition, $La_{0.40}Sr_{0.1}Ag_{0.48}Co_{0.93}O_3$, displays similar catalytic performances than those of a Pt film, for CO oxidation. In addition, this porous coating is also active for CO and NO oxidation in a simulated Diesel exhaust gas mixture, demonstrating the promising catalytic properties of such nanostructured thin sputtered perovskite films.

## 2. Results

### 2.1. Preparation and Characterization of LaCoO₃ Catalytic Coatings

Layers of $LaCoO_3$ were synthetized on alumina substrates by using the reactive magnetron sputtering preparation method. Depositions have been performed in a reactive mode, mixing oxygen and argon in the chamber. To achieve the suitable La/Co ratio, the discharge current applied to the Co target was adjusted while maintaining a constant current of 1 A on the La target (Table 1). As expected, the La/Co ratio, estimated by Energy Dispersive Spectroscopy (EDS), decreases with the current dissipated on the Co target (Figure 1). We determined that a current intensity of 0.3 A is

required to reach the targeted La/Co ratio. The diffractogram recorded on this as-deposited coating (Figure 2) shows that the layer is amorphous as no peak is detected. The difference between the radii of $La^{3+}$ (136 pm) and $Co^{3+}$ cations (72 pm) suggests an amorphous as-deposited coating due to steric effects as predicted by the confusion principle [33–35]. Then, a calcination step was performed at different temperatures (from 100 °C to 500 °C) for 2 h in air. X-Ray Diffraction (XRD) patterns (Figure 2), obtained under a Bragg-Brentano configuration (θ/2θ), evidence that the oxide coating crystallizes from 500 °C as a cubic perovskite phase (JCPDS 01-075-0279). This temperature is approximately 100 °C lower than those reported by H. Seim et al. [20] and H.J. Hwang et al. [36] on $LaCoO_3$ films prepared by PLD and sol-gel methods, respectively. This demonstrates that the reactive magnetron sputtering technique is efficient for preparing crystallized lanthanum cobaltite films at rather low temperatures.

**Table 1.** Sputtering parameters used for $LaCoO_3$ coating.

| Target | Intensity (A) | Pulse Frequency (kHz) | Dead Time (Toff μs) | Ar Flow Rate (sccm) * | O$_2$ Flow Rate (sccm) * | Total Pressure (Pa) | Draw Distance (mm) | Sputtering Time (h) |
|---|---|---|---|---|---|---|---|---|
| La | 1 | 50 | 5 | 50 | 20 | 1.5 | 45 | 3 |
| Co | 0.2 to 0.4 | | | | | | | |

* sccm = Standard Cubic Centimeter per Minutes.

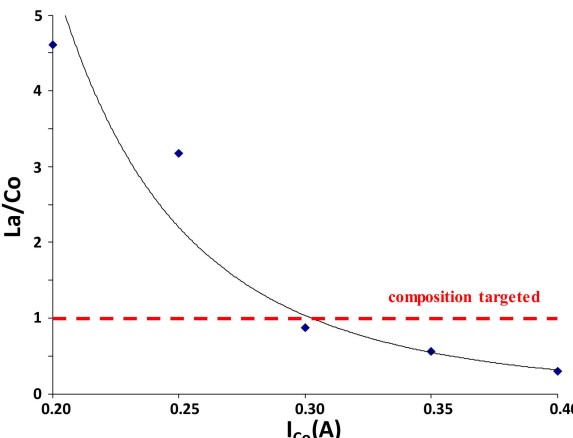

**Figure 1.** Evolution of the atomic La/Co ratio measured by Energy Dispersive Spectroscopy (EDS) as a function of the current dissipated on the Co target ($I_{La}$ = 1 A). The total pressure is 1.5 Pa.

This calcination step at 500 °C was performed for all coatings (Figure 3). The thin film with the highest La/Co ratio is not crystallized and only presents small peaks of $La_2O_3$ (JCDPS 00-005-0602), probably coming from a high excess of La. On the opposite, for larger atomic ratios, coatings are crystallised with various space groups of the perovskite-type structure. Indeed, the film is cubic (Pm3m space group) for a ratio of 0.87, exhibits a rhomboedric symmetry (R-3c space group) for a ratio of 0.56 and again becomes cubic for a ratio of 0.3.

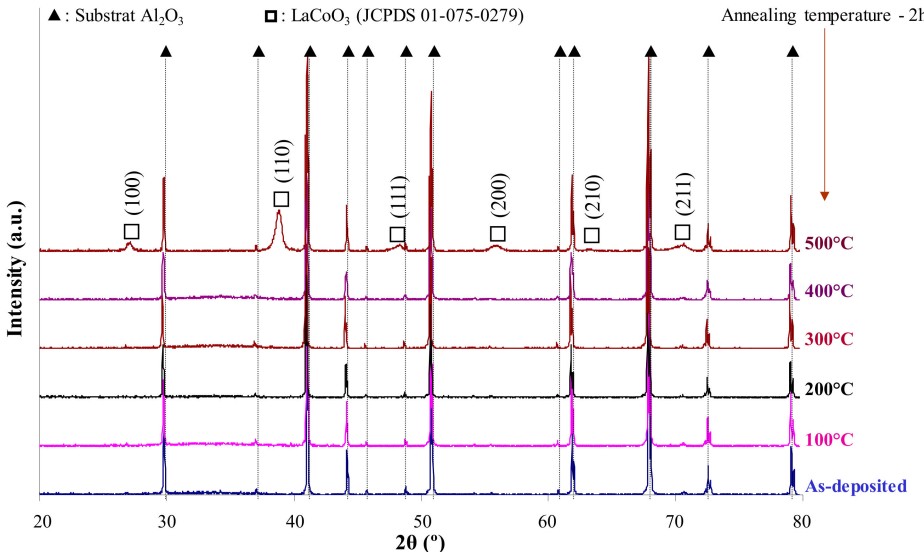

**Figure 2.** X-Ray diffractograms of LaCoO$_3$ sputtered films prepared with a current dissipated on the Co target of 0.3 A (La/Co = 0.87) after different post-calcination treatments during 2 h in air from 100 °C to 500 °C.

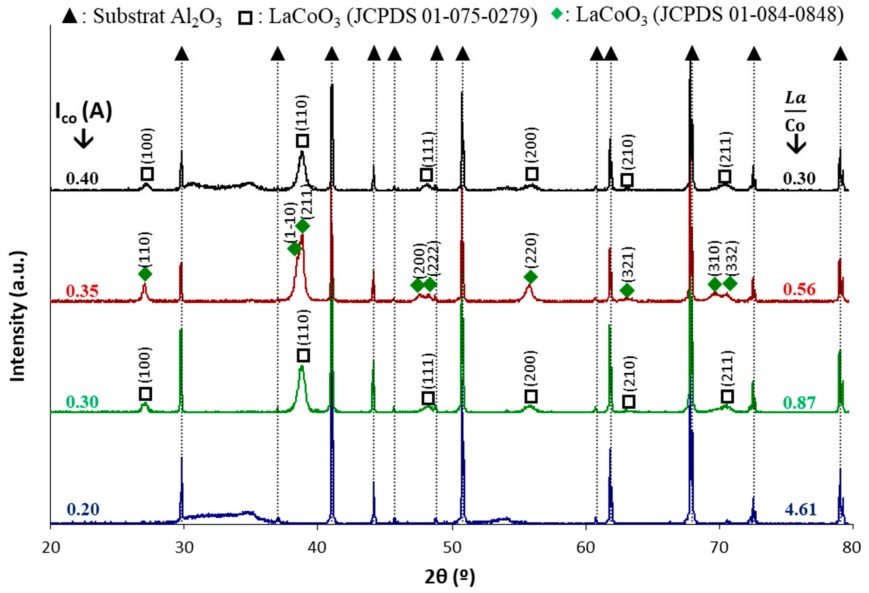

**Figure 3.** XRD patterns as a function of the atomic composition ratio after calcination treatment at 500 °C for 2 h under air.

The morphology of the LaCoO$_3$ coating with the suitable ratio (La/Co = 0.87) was observed by SEM. Figure 4 shows SEM images of the top view and the brittle cross section of the sample before and after the calcination step at 500 °C. The as-deposited coating (Figure 4a,c) covers the surface of the alumina substrate and follows its morphology. This film is quite dense, as shown in the cross-section image (Figure 4a) and adherent with a vitreous appearance characteristic of an amorphous material, in agreement with XRD (Figure 2) [37]. After calcination (Figure 4b), some cracks can be observed especially in the cross section (Figure 4d). The crystallized film is not sticking well with the alumina support and remains quite dense. The cracks were probably formed under stresses during the crystallization or the thermal treatment. In this latter case, the mismatch of thermal expansion coefficients between the perovskite film and the alumina substrate could be at the origin

of the delamination ($\alpha_{LaCoO3} \approx 20 \times 10^{-6}$ °C$^{-1}$ [38] and $\alpha_{Al2O3} \approx 7 \times 10^{-6}$ °C$^{-1}$ [39]) of the coating. The thickness of the annealed coating is around 1.5 µm (Figure 4c,d), resulting in a 500 nm.h$^{-1}$ deposition rate. This result is in agreement with the thickness measured by tactile profilometry.

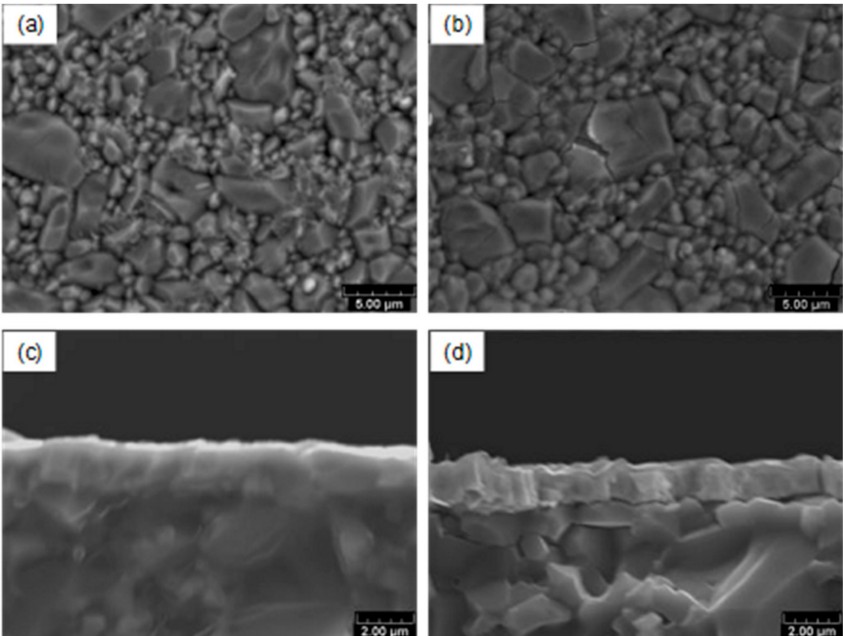

**Figure 4.** SEM images of the surface and the brittle cross section of the LaCoO$_3$ sample synthesized with I$_{Co}$ = 0.3 A and I$_{La}$ = 1 A: (**a,c**) as deposited coating and (**b,d**) after calcination treatment at 500 °C for 2 h under air.

### 2.2. Preparation and Characterization of La$_{1-x-y}$Sr$_x$Ag$_y$CoO$_{3-\alpha}$ (LSACO) Catalytic Coatings

The experimental parameters used for the synthesis of double substituted cobaltite (denoted as LSACO) coatings on alumina supports are reported in Table 2. The protocol is similar to that developed for the synthesis of pure lanthanum cobaltite (Ar flow rate = 50 sccm, O$_2$ flow rate = 20 sccm, total pressure = 1.5 Pa, draw distance = 45 mm) except the sputtering time which was extended to 4 h to achieve thicker films. The control of the substitution rates of Sr and Ag was achieved by tuning the intensity of the currents dissipated on the metallic targets (Table 2). We assumed that Sr$^{2+}$ and also Ag$^+$ cations will partially substitute La$^{3+}$ cations in A sites of the perovskite. This hypothesis is based on the similarity of the ionic radius of La$^{3+}$ (136 pm) and Ag$^+$ (128 pm) [40]. The Ag content in the perovskites was gradually enhanced by increasing the current applied to the Ag target (LSACO-1 to 4). This was counter-balanced by a progressive decay of the current dissipated to the La target. The Co content has been slightly increased in LSACO-5 compared with LSACO-4, while maintaining constant the Ag, La, and Sr contents (Table 2)

**Table 2.** Sputtering parameters used for *LSACO* coatings.

| Coating | I$_{La}$ (A) Pulse (kHz)/Toff (µs) | I$_{Sr}$ (A) Pulse (kHz)/Toff (µs) | I$_{Ag}$ (A) Pulse (kHz)/Toff (µs) | I$_{Co}$ (A) Pulse (kHz)/Toff (µs) |
|---|---|---|---|---|
| LSACO-1 | 1.2 50/5 | 1.2 350/1.4 | 0.008 0 | 0.35 50/5 |
| LSACO-2 | 0.8 50/5 | 1.1 350/1.4 | 0.009 0 | 0.35 50/5 |
| LSACO-3 | 0.75 50/5 | 1.1 350/1.4 | 0.01 0 | 0.35 50/5 |
| LSACO-4 | 0.6 50/5 | 1.1 350/1.4 | 0.013 0 | 0.35 50/5 |
| LSACO-5 | 0.6 50/5 | 1.1 350/1.4 | 0.013 0 | 0.4 50/5 |

The chemical composition of the as-prepared LSACO coatings was estimated by EDS analysis. Table 3 displays the variations of the atomic percent of each element, without considering the oxygen

level which is tricky to quantity by using EDS. The Sr and Ag contents inversely vary in LSACO-1, 2 and 3 while La and Co loadings are fairly stable. Let us note that the chemical composition of LSACO-2 and LSACO-3 is fairly similar with 11 at% of Ag and around 10 and 30 at% of Sr and La, respectively. The high current applied to the Ag target during the preparation of LSACO-4 and LSACO-5 leads to a significant increase in the Ag concentration and a concomitant drop of the La content. In LSACO-4 and LSACO-5, the Ag atomic percent reaches around 25% while the La level is approximately 20%. Surprisingly, the Co concentration decreases in LSACO-5 despite the largest applied current to the Co target. Except LSACO-5, the atomic ratio between cations located in A sites (theoretically $La^{3+}$, $Sr^{2+}$ and $Ag^+$) and $Co^{3+}$ ones located in B sites is close to the target of 1. The morphology of the as-deposited coatings (Figure 5) was found to drastically change with the chemical composition. The LSACO-1 coating only contains few perovskite clusters (Figure 5a, white spots at the top right) dispersed on the surface of the micrometric alumina grains (Figure 5, dark grey), despite of the 4 h deposition time. For LSACO-2 and LSACO-3, the number of perovskite clusters significantly increases on the surface of the alumina grains (white spots in Figure 5b,c). Interestingly, larger starfish shaped perovskites islands are growing on the substrate defects (cavity of alumina substrate or grain boundaries) showing arms like filaments. The morphology of the surface of LSACO-4 and LSACO-5 coatings (Figure 5d,e) are quite different. The alumina substrate is now fairly fully covered by a porous film with a wire-like morphology (Figure 5d,e). These wires have grown parallel to each other, leading void between each other, reaching a length of the order of 1 μm and a diameter of around 100 nm. Therefore, high Ag contents seem to strongly enhance the porosity and then the coverage of the films.

**Table 3.** Chemical composition determined by EDS of the as-prepared $La_{1-x-y}Sr_xAg_yCoO_{3-\alpha}$ (LSCAO) perovskite coatings.

| Coatings | La (at%) | Sr (at%) | Ag (at%) | Co (at%) | Chemical Formula |
|---|---|---|---|---|---|
| LSACO-1 | 29 ± 0.29 | 14 ± 0.14 | 7 ± 0.07 | 50 ± 0.5 | $La_{0.58}Sr_{0.28}Ag_{0.14}Co_1O_3$ |
| LSACO-2 | 29 ± 0.29 | 11 ± 0.11 | 11 ± 0.11 | 49 ± 0.5 | $La_{0.56}Sr_{0.23}Ag_{0.21}Co_{0.94}O_3$ |
| LSACO-3 | 32 ± 0.32 | 9 ± 0.1 | 11 ± 0.11 | 48 ± 0.5 | $La_{0.61}Sr_{0.18}Ag_{0.21}Co_{0.92}O_3$ |
| LSACO-4 | 20 ± 0.20 | 7 ± 0.1 | 25 ± 0.25 | 48 ± 0.5 | $La_{0.39}Sr_{0.13}Ag_{0.49}Co_{0.93}O_3$ |
| LSACO-5 | 21 ± 0.21 | 9 ± 0.1 | 28 ± 0.28 | 42 ± 0.4 | $La_{0.40}Sr_{0.20}Ag_{0.48}Co_{0.70}O_3$ |

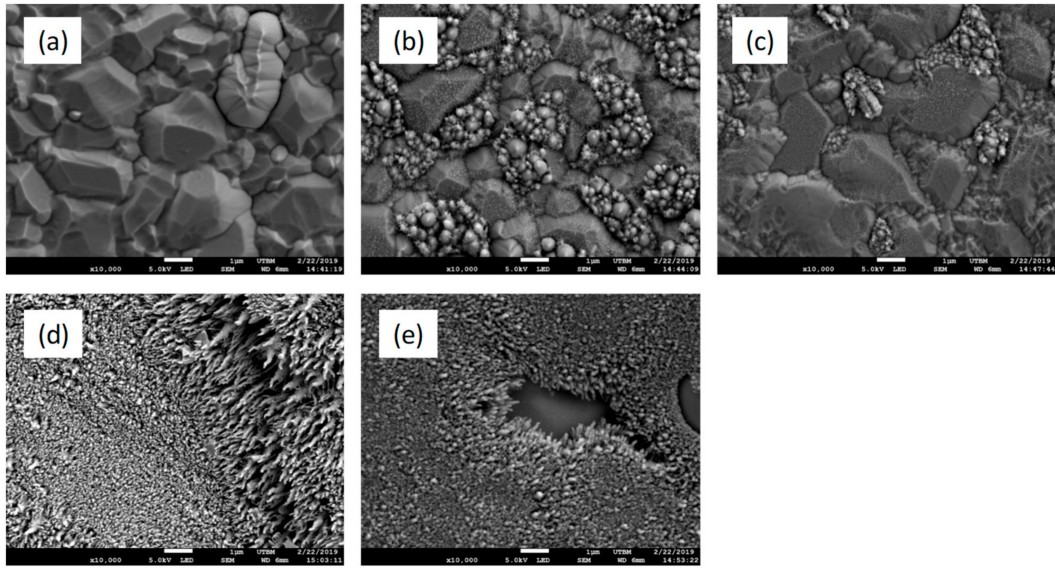

**Figure 5.** SEM images of the surface morphology of as-deposited LSACO coatings on alumina substrate: (**a**) LSACO-1, (**b**) LSACO-2, (**c**) LSACO-3, (**d**) LSACO-4, and (**e**) LSACO-5.

These series of LSACO coatings were annealed at 500 °C for 2 h in air. This calcination step was sufficient to crystallize all the films as a cubic perovskite phase (JCPDS 01-075-0279) whatever the composition. On the other hand, an additional XRD pattern at 44.3° on the diffractogramm of LSACO-4 and LSACO-5, corresponding to (2 0 0) planes of fcc metallic silver, proves the presence of metallic Ag, out of the perovskite structure. These results demonstrate the limited solubility of Ag in the perovskite in agreement with previous studies [32]. For high contents of Ag, reaching 25 at. %, i.e., around 10 at% in the overall oxide including the oxygen atoms, part of Ag is not incorporated into the perovskite structure. No XRD peaks corresponding to Ag° were observed for LSAC0-1, LSACO-2, and LSACO-3 (Figure 6), but the low content of Ag, below 5 at%, makes difficult their detection. Therefore, we cannot exclude the presence of metallic Ag on the surface of these samples.

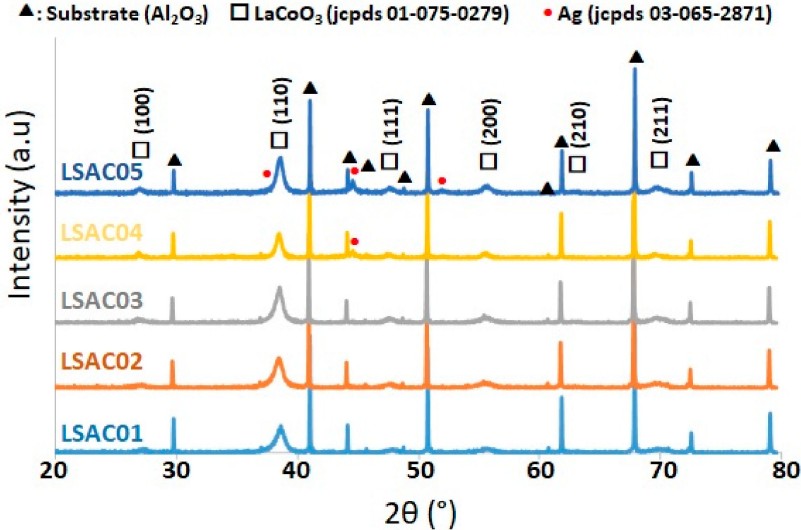

**Figure 6.** XRD patterns after the calcination treatment at 500 °C for 2 h under air of the different LSACO coatings.

Figure 7 displays the surface morphology of the calcinated films. The alumina substrates are cracked probably due to the thermal treatment. The LSACO-1 coating only contains isolated perovskites clusters on the surface of alumina grains. As before calcination, the morphology of LSACO-2 and LSACO-3 coatings are similar with perovskite islands mainly located in the cavity and interstices of the alumina substrate. Filaments that were present around of the perovskite clusters have disappeared. Coatings containing high Ag loadings (LSACO-4 and LSACO-5) are the only ones able to fully cover the alumina substrate with a porous wire-like morphology. These nanowires are less ordered than before calcination and interlock, then decreasing the porosity of the film. Ag particles (Figure 7e) can be observed on the surface of LSACO-5 (white particles), confirming that a part of Ag was not incorporated into the perovkiste structure.

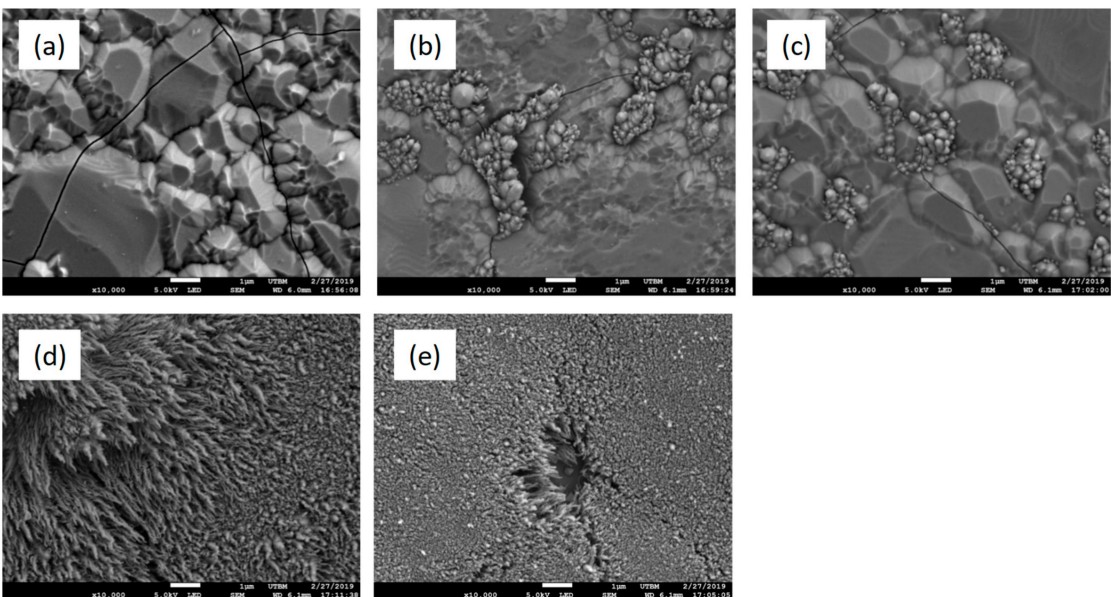

**Figure 7.** SEM images of the surface morphology of calcined LSACO coatings on alumina substrate: (**a**) LSACO-1, (**b**) LSACO-2, (**c**) LSACO-3, (**d**) LSACO-4, and (**e**) LSACO-5.

## 2.3. Catalytic Performances of the Perovskite Coatings

### 2.3.1. Catalytic Performances for CO Oxidation

The catalytic activity of LSACO-2, LSACO-4, and LSACO-5 coatings, containing respectively 11, 25 and 28 at% of Ag, was measured for CO oxidation during heating ramps up to 400 °C. We have selected LSACO-4 and LSACO-5 as these films exhibit a porous and wire-like morphology. We have also tested the catalytic performances for CO oxidation of pure lanthanum cobaltite coatings. These samples have shown no activity below 500 °C in good agreement with their dense morphology. The catalyst LSACO-2 was also tested for comparison as a non-porous and non-covering representative layer. In addition, we have also tested, for comparison, a Pt coating (3 μm thick, 5 mg Pt/cm$^2$) prepared by spray-painting and annealed at 500 °C. Figure 8 displays the Light-off (LO) curves of the different catalytic coatings for CO oxidation. The results show that the catalytic performances increase with the silver content in the film. Values of T20, temperature at 20% conversion, significantly decrease from 280 to 215 °C when increasing the Ag content. Furthermore, the catalytic performances of LSACO-5 are rather close to those of the Pt coating up to 220 °C. This underlines the promising catalytic activity of the sputtered Ag-doped perovskite coating.

Two successive LO up to 400 °C have been performed on LSACO-4 (Figure 9a). The value of T20 increases from 215 to 242 °C during the second LO. This indicates that a modification of the coating morphology during the first LO, most probably due to the Ag particles sintering. Nevertheless, the onset temperature slightly decreases from 175 to 150 °C (Figure 9a). Similar catalytic experiments have been carried out on LSACO-5 (Figure 9b) without any significant modification of the catalytic performances between the two successive LO. This indicates a better stability of the morphology of LSACO-5 in the presence of the reactive mixture up to 400°C compared with LSACO-4. XPS measurements performed after catalytic tests evidence a La surface seggregation with a concommitant drop of the Co concentration. The two catalytic coatings show a similar Ag surface atomic concentration, i.e., around 10 at%, which is far lower from the loading estimated by EDS before the catalytic tests (Table 3). The Ag3d XPS peaks (Figure 10) show a binding energy of Ag3d5/2 at 367.7 eV, attributed to metallic silver. The catalytic properties during the second LO of the two catalytic perovskite coatings are fairly similar, in good agreement with an equivalent Ag surface concentration (Table 4). This confirms the direct link between the surface Ag concentration and the catalytic activity for CO oxidation.

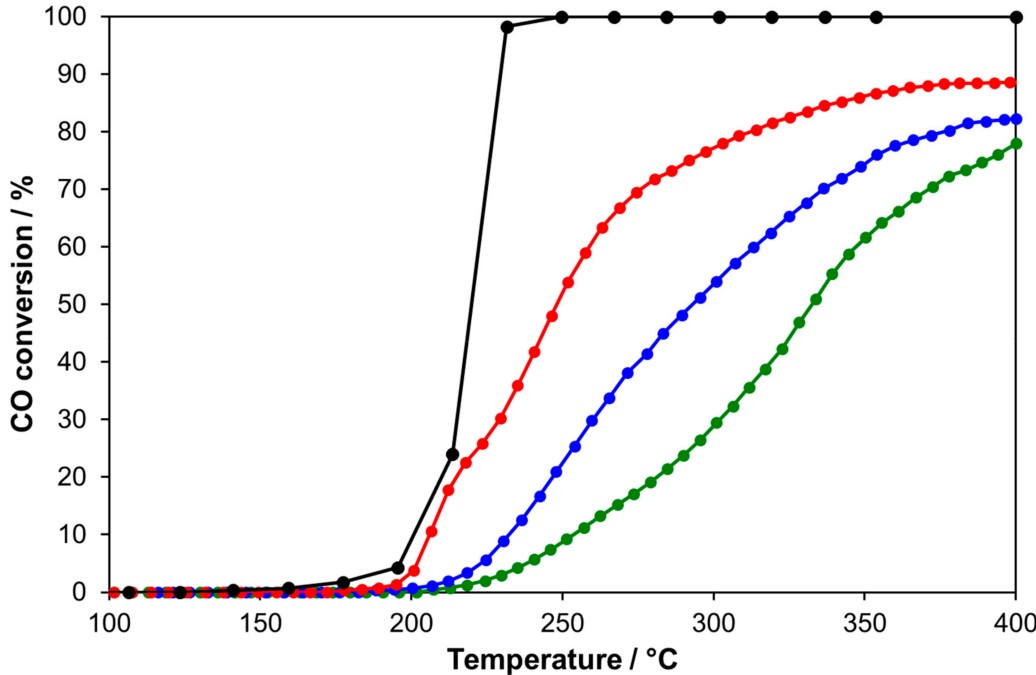

**Figure 8.** Light-off curves for CO oxidation recorded on the sputtered perovskite coatings and on Pt film. Reactive mixture: $CO/O_2$ = 3000 ppm/3%. Overall flow = 3.6 L/h.

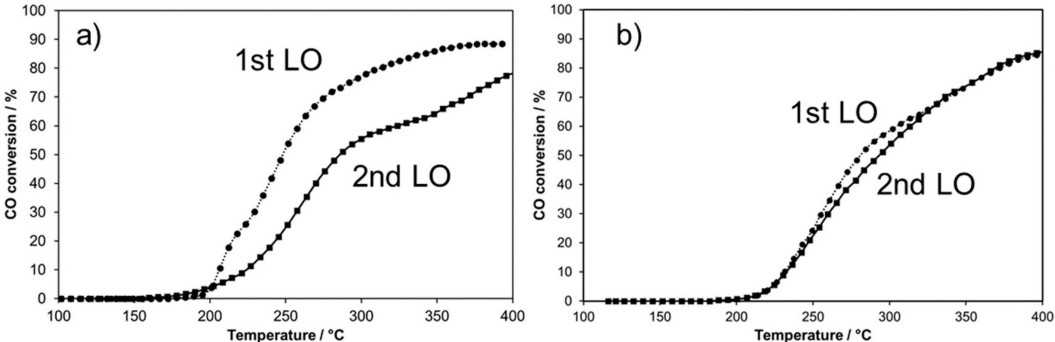

**Figure 9.** Two successive LO recorded on (**a**) LSACO-4 and (**b**) LSACO-5. Reactive mixture: $CO/O_2$ = 3000 ppm/3%. Overall flow = 3.6 L/h.

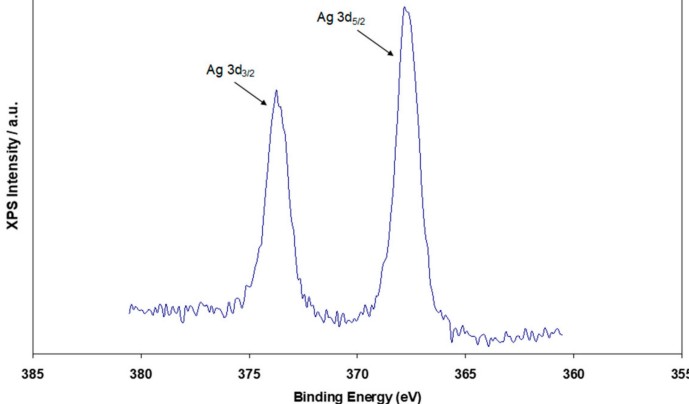

**Figure 10.** Ag3d XPS spectrum of LSAC-5.

**Table 4.** Surface composition determined by XPS of LSACO-4 and LSACO-5 after catalytic tests.

| Sample | La at% | Sr at% | Ag at% | Co at% |
|--------|--------|--------|--------|--------|
| LSACO-4 | 47 | 11 | 12 | 30 |
| LSACO-5 | 51 | 13 | 10 | 26 |

### 2.3.2. Catalytic Performances in a Model Lean Diesel Exhaust Gas

The catalytic performances of LSACO-4 were explored in a model lean diesel exhaust gas mixture containing 8% $O_2$, 950 ppm CO, 270 ppm NO, 1000 ppm $C_3H_8$ and 10% $H_2O$. The LO was recorded up to 375 °C to investigate the low temperature activity. Figure 11 shows the ability of the catalytic coating to oxidize CO and NO into $CO_2$ and $NO_2$, respectively. Below 350 °C, we did not observed any propane oxidation. However, the catalytic coating shows a remarkable activity for NO oxidation from around 225 °C. The NO conversion achieves 30% at 375 °C. CO conversion starts from 250 °C but seems to reach a maximum at only 18% from 350 °C. These results show that, despite a very low mass of catalyst and a complex mixture including the inhibiting effect of $H_2O$, the perovkite catalytic coating can be active at low temperature for low temperature NO and CO oxidation.

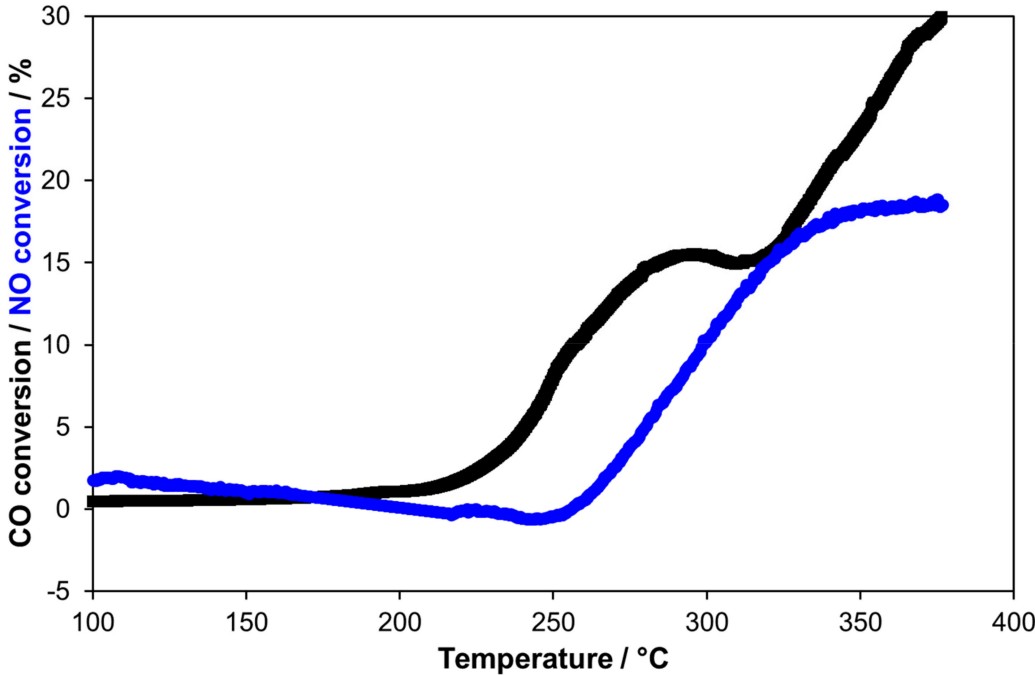

**Figure 11.** Variation of CO and NO conversion as a function of temperature on LSACO-4. Reactive mixture: CO/$O_2$/NO/$C_3H_8$/$H_2O$: 950 ppm/8%/270 ppm/1000 ppm/10%. Overall flow: 3.6 L/h.

## 3. Materials and Methods

### 3.1. Preparation of the Catalytic Coatings

$LaCoO_3$ perovskite coatings were synthesized by magnetron sputtering from two metallic targets of La (kurt J. Lekser, purity 99.9%, Hastings, England) and Co (kurt J. Lekser, purity 99.9%). The Alcatel SCM 604 reactor, described elsewhere [33], was a 90-l sputtering chamber where a pressure of $10^{-4}$ Pa was maintained by a primary pump assisted by a molecular turbo pump. The La metallic target (thickness = 3 mm; diameter = 50 mm) and the Co one (thickness = 1 mm; diameter = 50 mm) were attached to two magnetrons distant from 120 mm and connected to a pulsed direct current generator (PINACLE+ purchased by advance energy). The distance between the two targets and the substrate holder (DT-S) can be independently modified. The draw distance between the targets and the alumina

substrate was adjusted to 45 mm (Table 1). The targets are powered with pulsed currents to avoid any electric micro-arcs that can damage the quality of the films [41]. To avoid any accumulation of positive charge (Ar+) on the targets, the current is periodically stopped for very short dead time of 5 μS. Gases were introduced with mass-flow controllers (Brooks, 5850 SLA, Hatfield, PA, USA) and the total working pressure was measured with a MKS Baratron gauge. Films were deposited on dense alumina pellets (Keral 99, diameter = 16 mm, thickness = 0.60 mm purchased by Kerafol Gmbh, (Koppe Platz 1, Eschenbach i. d. Opf). The synthesis of doped perovskites with Sr and Ag was performed with the addition of two magnetrons into the sputtering chamber and metallic targets of Sr and Ag (thickness = 3 mm diameter = 50 mm, kurt J. Lekser, purity 99.9%). The Sr and Ag targets were powered by a two pulsed direct current generator (PINACLE+ and MDX500 both purchased by advance energy, Metzingen, Germany).

### 3.2. Characterizations of the Catalytic Coatings

The crystal structure of the coatings was determined thanks to a BRUKER D8 focus X-Ray diffractometer (Co K$\alpha$1+$\alpha$2 radiations, in Bragg Brentano configuration and equipped with a LynxEye linear detector ( Bruker, Billerica, MA, USA). XRD patterns were collected under air during 10 min in the (20–80°) scattering angle range by steps of 0.019°. The surface and the morphology of films were observed with a scanning electron microscope (JEOL JSM 7800F, Akishima, Japan) equipped with an EDS detector allowing the estimation of the chemical composition of samples. The observation of brittle cross section by SEM led also to the determination of the thickness of films. Coating thickness was also determined by the step method with an Altysurf profilometer produced by Altimet society (Marin, France) equipped with tungsten micro force probe inductive allowing an accuracy of about 20 nm. Before each measurement, the calibration of the experimental device was realized with a reference sample number 787569 accredited by CETIM organization.

XPS spectra were recorded for each catalysts on an AXIS Ultra DLD from Kratos Analytical (Manchester, UK) using a monochromatized Al X-ray source (h$v$ = 1486.6 eV) between 0 and 1200 eV with a pass energy of 40 eV. Sample were pretreated at 200 °C in He before measurements to clean the surface. Peaks were referenced using C1S peak of carbon (BE = 284.6 eV).

### 3.3. Catalytic Activity Measurements

A quartz reactor was operated under continuous flowing conditions at atmospheric pressure. The samples were placed on a fritted quartz, 18 mm in diameter, with the catalytic coating side facing the fritted quartz [42]. Gases were mixed by using mass-flow controllers (Brooks) to generate the different reactive mixtures. $H_2O$ vapor was introduced using an atmospheric pressure Pyrex saturator heated at 46 °C. Reactants and products were analyzed using a gas micro-chromatograph (SRA 3000 equipped with two TCD detectors, a molecular sieve and a Porapak Q column for $O_2$, CO, $C_3H_6$, and $CO_2$ analysis) and a $CO_2$ Infra-Red analyzer (Horiba VA 3000, Horiba Europe Gmbh, Leichlingen, Germany). NO and $NO_2$ concentrations were measured with IR and UV online analyzers (EMERSON NGA2000).

Reactants were Air Liquide certified standards of NO in He (8000 ppm), $C_3H_8$ in He (8005 ppm), CO in He (1%), $O_2$ (99.999%), which could be further diluted in (99.999%). The carbon and nitrogen balance closure was found to be within 2%.

## 4. Conclusions

Different compositions of $La_{1-x-y}Sr_xAg_yCoO_{3-\alpha}$ ($x$ = 0.13–0.28, $y$ = 0.14–0.48) doped perovskites were synthetized as thin coatings deposited on alumina disks with the cathodic co-sputtering magnetron method in reactive conditions. The control of different parameters during the sputtering process can tune the morphology of the catalytic films. In particular, we found that the incorporation of high Ag loadings can generate covering films with a porous wire-like morphology showing good catalytic activity for CO oxidation. The most active composition, $La_{0.40}Sr_{0.1}Ag_{0.48}Co_{0.93}O_3$, displays similar

catalytic performances than those of a Pt film. In addition, this porous coating is also efficient for CO and NO oxidation in a simulated Diesel exhaust gas mixture, demonstrating the promising catalytic properties of such nanostructured thin sputtered perovskite films.

**Author Contributions:** Investigation, M.A.P.Y. and L.L.; data curation, P.B. and P.V.; writing—original draft preparation, M.A.P.Y, P.B., and P.V.; writing—review and editing, P.B. and P.V.; formal analysis: A.B., P.V. supervision, A.B., P.B. and P.V.; project administration, P.B.; funding acquisition, P.B., P.V. and A.B.

**Funding:** The research was funded by "ADEME" in the frame of the ADIABACAT project and "Pays de Montbéliard Agglomération".

**Conflicts of Interest:** The authors declare no conflict of interest. The founding sponsors had no role in the design of the study; in the collection, analyses, or interpretation of data; in the writing of the manuscript, or in the decision to publish the results.

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
