# Peer review of "Catalytic Properties of Double Substituted Lanthanum Cobaltite Nanostructured Coatings Prepared by Reactive Magnetron Sputtering"

_catalysts, doi:10.3390/catal9040381_

Round 1

Reviewer 1 Report

Report on the submitted research article Catalysts - 484785

            The manuscript describes the catalytic properties of several double substituted lanthanum cobaltite nanostructured coatings prepared by reactive magnetron sputtering. The authors investigate the morphology, the atomic ratio and structure, and the catalytic performance of some of the produced coatings, concluding that the surface concentration of Ag plays decisive role to the catalytic activity for CO oxidation. Although the results look original and very interesting, unfortunately the manuscript is not very carefully written (at least in some parts). For example there is confusion as far as it concerns the numbers of some figures. Therefore, in order the manuscript to be accepted, I suggest to the authors to improve the text, thus presenting their important results more clearly and with more lucidity. The following lists may help in this direction.

Majors

1)      Results, line 82: Why the suitable La/Co atomic ratio is 0.5?

2)      Table 1: Please explain the flow rate units (sccm).

3)      Line 143: Where the values of ionic radius were taken from? Furthermore, I see no ionic radius for Sr.

4)      Lines 161-162: The authors discuss about A and B sites. It will be helpful to show these sites in the unit cell of the perovskite, by drawing a picture of the atomic structure.

5)      Line 171: There are no Figures 6d and e. Do you mean 5 d and e?

6)      Line 186: Fig.5 shows no XRD. I guess the authors mean Fig. 6.

7)      Figs 5b and c, in comparison with Figs 7b and c: For the as-deposited LSACO-2 and LSACO-3 samples, the authors speak about perovskite clusters with a star shape morphology. For the calcined LSACO-2 and 193 LSACO-3 coatings (Figs 7b and c) the authors mention that the filaments of the perovskite clusters disappear. What do they mean? I see no filaments in Figs 5b and c. Please explain.

8)      Line 210:The sample LSACO-5 is 28% or 27% at Ag? I think the chemical formula in Table 3 gives 27%.

9)      Line 216: What T20 values are? Please explain.

10)  Figure 9: Why the two successive LO curves differ from each other in the case of LSACO-4, while it doesn’t happen the same in the case of LSACO-5?

Minors

1)      Abstract: Please write” high Ag.”

2)      Results, page 3: Please give the definitions of XRD and EDS, wherever they appear for first time.

3)      Reference 14: The title is wrong. Please correct.

4)      Reference 34: The authors are at the end.

5)      Reference 35: The title is missing.

6)      Line 109: Dissipated?

7)      Line 135: Write in italic the title of the 2.2. section.

8)      Line 191: “The alumina substrates are cracked…”. Please correct the grammar.

Author Response

Dear reviewer,

we thank you for your comments, and we have answered all your questions in the word file. We also completed the article

Best regards

Pascal

Reviewer 2 Report

The paper entitled “Catalytic properties of double substituted lanthanum cobaltite nanostructured coatings prepared by reactive magnetron sputtering” is devoted to an alternative approach for synthesis of perovskite coatings active in CO oxidation and non-containing noble metals. The paper is quite interesting and presents new results in the studied field. I can recommend it for publication in Catalysts after minor revision. The comments are following.

1.      The aim of the study should be corrected. Initially it was stated that one of the advantages of the proposed catalytic coatings is replacement of the noble metals. As well known, the noble metals are silver, gold, platinum, rhodium, iridium, palladium, ruthenium and osmium. How silver-substituted perovskite can replace catalysts containing noble metals?

2.      It is stated few times that “development of catalytic films may optimize the surface/bulk ratio to save raw materials”. What is meant? It should be clarified.

3.      One of the additional advantages of the developed coatings is considered to be oxidation of NO, especially when using a simulated diesel exhaust gases. It is quite amazing, since, to the best of my knowledge (I’m dealing with diesel exhaust gases since 2004), NOx problem is solved using SCR technology. No NO oxidation is required for characterization of diesel oxidation catalysts.

4.      There is confusion with La/Co ratio. A lot of writing variations are used: La/Co atomic ratio; (at.% Co)/(at.% La + at.%Co); [(Co) /(Co+La)] atomic ratios; Co/(La+Co) ratio; Co/(Co+La) ratio; ([(Co) /(Co+La)] atomic ratio.

5.      EDS or EDX? Please, used one abbreviation and define it.

6.      All “hour” should be shortened to “h”.

7.      Line 185: “No XRD patterns corresponding to Ag° were observed …”. Patterns cannot be observed only in the case if you didn’t register them. In the present case you are talking about the reflexes.

8.      Quite often authors have used both Kelvin degree and degree Celsius.

9.      Light-off curves for CO oxidation (Figs. 8, 9) should be given for full temperature range and up to 100% of conversion. In the present form they are unacceptable.

10.  Why red curve in Fig. 8 is stopped at ~220 °C?

11.  Line 227: “This indicates a better stability of the catalytic activity of LSACO-5…” – it is wrong belief. It indicates nothing, since the final temperature of the catalytic test wasn’t high enough.

12.  There are two Figs. 10.

13.  Title of subsection 2.3.2 should be improved.

14.  Title of 3.1 – change to “Preparation of the catalytic coatings”

15.  Line 272: “introduced with mass flowmeters” – may be “mass-flow controllers”?

16.  Line 294: “scanning energy was 40 eV for every region” – what is meant? Data for just one region are presented.

17.  It is not described in the 3.3 section how the NO concentration was measured/calculated.

18.  References 26, 27, and 34 are out of format.

19.  English of the manuscript should be carefully checked and corrected. The text contains a lot of mistakes and misprints. Some of them are presented below.

19.1.                   Abstract: “High Ah loadings” – misprint

19.2.                   Abstract: “a porous like-wire morphology” – change to “a porous wire-like morphology”. The same in lines 197, 211 and 315.

19.3.                   All over the text: cristallizes (p. 3, line 91); cirstallized (p. 3, line 95); cristallized (p. 5, line 110); crystallised (p. 5, line 112)

19.4.                   P. 3, line 93: “and al.” change to “et al.”

19.5.                   “charcteritics” (p. 5, line 110) – misprint

19.6.                   In general, there is troublesome sentence in lines 110-111 (p. 5).

19.7.                   Figure 3 caption: “XRD spectra” change to “XRD patterns” or “X-ray diffractograms”

19.8.                   Figure 4 caption: “synthesised” – misprint

19.9.                   Line 137: “on Table 2” change to “in Table 2”

19.10.               Line 146: “sliglty” – misprint

19.11.               Line 154: “… contents … varies”

19.12.               Line 156: “laodings” – misprint

19.13.               Line 160: “Surprinsingly,” – misprint

19.14.               Line 171: “wires have growth parallel to each other” – may be “wires have grown parallel to each other”?

19.15.               Line 179: “jcpds” change to “JCPDS”

19.16.               Line 184: “part of Ag in not incorporated”

19.17.               Line 224: “sligtly” – misprint

19.18.               Lines 158, 229: “concomittant drop”, “concommitant drop” – misprints

19.19.               Figure 9 caption: “…on a) LSACO-4And b) LSACO-5.”

19.20.               Line 291: Missing full stop

Author Response

(The authors gave the same response as above.)

Reviewer 3 Report

Catalytic properties of double substituted lanthanum cobaltite nanostructured coatings prepared by reactive magnetron sputtering

by M. Arab Pour Yazdi, L. Lizarraga, P. Vernoux, A. Billard, P. Briois

The manuscript presents the synthesis of thin catalytic coating of lanthanum perovskite, doped with Sr and Ag, deposited on alumina disks, through the chatodic co-sputtering magnetron method in reactive conditions. The prepared materials were characterized by XRD, XPS, SEM and EDX analysis, while the catalytic performances were evaluated in the in the processes of CO oxidation and of diesel exhaust gases treatment. The synthesis method adopted is quite innovative, especially in such applications, and the doped perovskites exhibit a reactivity comparable to that of noble metals catalysts. Then, in the opinion of this reviewer, the manuscript is globally well-written and the results are properly discussed, although some minor changes and/or supplementary information are required to improve the quality of the work. In particular:

1.     The words “calcination” and “annealing” for the treatment in air at different temperatures, are used as synonyms.  I suggest to use only “calcination”, which is more correct in this context.

2.     The unit of measure for temperature should be uniformed and reported all in K or °C.

3.     English language should be revised and typos corrected. Ex. “quantity = quantify” (line 154), “concomittant = concomitant” (line 158), “echantillons = sample” (line 238).

4.     Line 108 “This calcination step at 573K…”, probably the correct temperature is 773K.

5.     Line 171 “Figure 6d and e” is “Figure 5.d and e”. Line 186 “Fig. 5” is “Figure 6”.

6.     The catalytic results in the CO oxidation should be reported, if possible, also in term of reaction rates, taking into account the real weight of catalyst used for each test (i.e. molCO2/gcat·h). Moreover, in order to better emphasize the relevance of the work, a detailed comparison with experimental results reported in literature should be added, providing upgraded literature analysis (ex. Journal of Rare Earths, https://doi.org/10.1016/j.jre.2018.11.011; Applied Catalysis B: Environmental 210 (2017) 14–22; Topics in Catalysis https://doi.org/10.1007/s11244-018-1113-0).

7.     For a better comparison, catalytic data in the CO oxidation, obtained with the pure LaCoO3 coating should be added in Section 2.3.1 “Catalytic Performances for CO oxidation”. Moreover, the effect of Sr addition should be better clarified and discussed.

8.     The surface composition of fresh LSACO-4 and LSACO-5 catalysts should be reported in Table 4 and discussed/compared with that of used systems.

Author Response

(The authors gave the same response as above.)
